# The Impact of COVID-19 and the Practical Importance of Vaccinations and Nirmatrelvir/Ritonavir for Patients with Cardiovascular Disease

**DOI:** 10.3390/vaccines13060554

**Published:** 2025-05-23

**Authors:** Marcin Wełnicki, Artur Mamcarz, Ernest Kuchar, Przemysław Mitkowski, Jerzy Jaroszewicz, Krzysztof Tomasiewicz, Mariusz Gąsior, Przemysław Leszek, Karol Adam Kamiński, Jacek Wysocki

**Affiliations:** 13rd Department of Internal Medicine and Cardiology, Medical University of Warsaw, 02-091 Warszawa, Poland; artur.mamcarz@wum.edu.pl; 2Department of Pediatrics with Clinical Assessment Unit, Medical University of Warsaw, 02-091 Warszawa, Poland; ernest.kuchar@wum.edu.pl; 31st Department of Cardiology, Poznan University of Medical Sciences, 60-512 Poznan, Poland; przemyslaw.mitkowski@usk.poznan.pl; 4Department of Infectious Disease and Hepatology, Faculty of Medical Sciences in Zabrze, Medical University of Silesia, 40-635 Katowice, Poland; jjaroszewicz@sum.edu.pl; 5Chair and Department of Infectious Diseases, The Medical University in Lublin, 20-059 Lublin, Poland; krzysztof.tomasiewicz21@gmail.com; 63rd Department of Cardiology, Faculty of Medical Sciences in Zabrze, Medical University of Silesia, 40-635 Katowice, Poland; m.gasior@op.pl; 7Department of Heart Failure and Transplantation Medicine, Cardinal Stefan Wyszynski Institute of Cardiology in Warsaw, 01-938 Warsaw, Poland; p.leszek@ikard.pl; 8Department of Population Medicine and Lifestyle Diseases Prevention, Medical University of Bialystok, 15-269 Bialystok, Poland; karol.kaminski@umb.edu.pl; 9Chair and Department of Health Prophylaxis, Medical University of Poznan, 61-701 Poznań, Poland; jawysocki@pro.onet.pl

**Keywords:** vaccine safety, pharmacotherapy of COVID-19, drug interactions

## Abstract

The COVID-19 pandemic posed a huge challenge to global health systems. In addition to searching for effective methods of treating and preventing infection with a new pathogen, we could once again observe that severe respiratory infection and its complications can be become a challenging problem for cardiac patients. Empirical observations are fully confirmed by the results of clinical trials. Patients with risk factors and already diagnosed with cardiovascular diseases are particularly exposed to the severe course of COVID-19, including death. That is why we consider it so important to promote vaccinations against COVID-19 as a safe and effective method of preventing serious infections in this special group of patients, in accordance with the updated recommendations of relevant experts. If an infection is detected, depending on its form and the risk of hospitalization, there are also several antiviral treatment strategies. Nirmatrelvir/ritonavir therapy is particularly effective in selected patient groups, but its use requires analysis of the cardiac pharmacotherapy regimen in the context of potentially significant drug interactions.

## 1. Background

On 5 May 2023, the director of the World Health Organization (WHO) announced that COVID-19 no longer constituted a global health threat. Colloquially, this day has come to be referenced as the “end of the COVID-19 pandemic”. In Poland, the official status of COVID-19 as an epidemiological threat ended on 1 July 2023. After years of fighting the new virus, a period marked by a disorganized healthcare sector and ad hoc solutions, it seemed that science had bested the disease. In record time, a vaccination against the new virus had been developed and successfully distributed. Research was also carried out under essentially combat conditions, and the effectiveness of known and new antiviral drugs was assessed, with various outcomes, including some clinical trials that ended with official registrations. The end of the pandemic was expected to signal that despite the continued presence of the new pathogen in our environment, we had acquired the tools, knowledge, and experience necessary to effectively fight it. The ongoing aftermath has demonstrated instead that the problem of COVID-19 persists and that chronically ill patients, including people with cardiovascular diseases, remain particularly at risk of the negative consequences of COVID-19. In an era of compartmentalized medical specialties, the COVID-19 pandemic is a reminder and underlines the validity of looking at a patient’s health holistically. The aim of this narrative review is to highlight effective prevention and treatment options for COVID-19 in patients with cardiovascular conditions. In an interdisciplinary group of representatives from various specialties, internal medicine, cardiology, infectious diseases, virology, and clinical pharmacology, we have developed a summary of current data on the impact of COVID-19 in patients with cardiovascular disease, as well as the efficacy and safety of vaccination and current pharmacotherapy options. The aim of this narrative review is to provide arguments supported by scientific evidence to justify the promotion of COVID-19 vaccination by cardiologists, as well as to critically present the possibility of pharmacotherapy in the event of infection

## 2. The Impact of COVID-19 on Death Rates

The strongest risk factor for death in the course of COVID-19 is older age. The in-hospital mortality rate among patients treated for COVID-19 was less than 10% for those under the age of 50, but during 2020 and 2021, rates increased with age after the fifth decade of life, reaching approximately 50% in men aged ≥ 85 years [1]. In materials developed by the US Centers for Disease Control and Prevention, based on meta-analyses, other confirmed risk factors for a severe course and death due to COVID-19 include cerebrovascular diseases, chronic kidney disease, type 1 and type 2 diabetes, and cardiovascular system conditions [1]. Data published in 2022 by the National Institute of Public Health PZH—National Research Institute (NIZH-PZH PIB) identified the highest mortality rate among patients treated for cardiovascular and respiratory diseases and other infectious diseases. The data also showed that mortality from COVID-19 increased greatly with age and was higher among men than women. In 2021, COVID-19 was the second most common cause of death among people in Poland, and 50% of those who died from COVID-19 were over the age of 80 years among women and over the age of 74 (in 2020) or 72 (in 2021) years among men [2]. In 2020–2021, an estimated minimum of 130,000 people in Poland died because of COVID-19. This number refers only to confirmed cases and does not take into account delayed deaths from long-term cardiovascular complications of infection or deaths related to restrictions on access to health services, including for oncological diagnostics. Observations over the initial years of COVID-19 are just a hint of the costs of the pandemic that are expected to accumulate in the coming years. According to data from NIZH-PZH PIB, in the first year of the pandemic, mortality related to circulatory system diseases in Poland was higher than expected by 8.8%, and in the following year, it was higher by 12.6% [2]. The number of hospitalizations for heart failure decreased, but the mortality rate among patients diagnosed with heart failure increased. Deterioration of care for cardiac patients was observed in inpatient and outpatient services, and the decline in the number of hospitalizations for cardiovascular diseases fluctuated with the severity of COVID-19 cases [1]. Research clearly shows that cardiovascular diseases and their classic risk factors increase the risk of illness and a severe course of COVID-19 [3,4,5,6,7,8,9,10,11,12,13,14,15,16].

## 3. Cardiovascular Diseases and the Risk of Illness and a Severe Course of COVID-19

To date, a number of studies have established the relationship between cardiovascular diseases or their risk factors and the risk of disease and a severe course with SARS-CoV-2 infection. Bae et al. conducted a meta-analysis of data from 51 studies with over 48,000 patients with confirmed coronavirus infection [3]. The authors found that the risk of severe COVID-19 or death was clearly higher with hypertension (odds ratio [OR] 2.50; 95% confidence interval [CI] 2.15–2.90), diabetes (OR 2.25; 95% CI 1.89–2.69), and previously diagnosed cardiovascular diseases (OR 3.11; 95% CI 2.55–3.79) [3]. Risk values for individual indicators also depended on age. Arterial hypertension occurred in approximately 26% of the overall study population but in 47% of the subgroup aged ≥ 60 years. When hypertension was considered as a prognostic factor for the course of COVID-19, the disease course was worse in patients under the age of 50 years (for severe course or death due to COVID-19, OR 3.49; 95% CI 2.24–5.45). A similar relationship was observed in the case of coexistence of diabetes or cardiovascular diseases (Table 1) [3].

Cannata et al. also obtained results of considerable interest in their analysis of in-hospital mortality among patients hospitalized for heart failure, acute coronary syndromes, and ST-elevation myocardial infarction, comparing the period before the COVID-19 pandemic to the pandemic era [4]. The analysis included 15 studies involving more than 27,000 patients. It should be emphasized that the primary cause of hospitalization was not COVID-19 [4], so the authors assessed the indirect impact of the pandemic on the cardiovascular prognosis of cardiac patients. They found that in-hospital mortality was 5.7% (1026 deaths) in the pre-pandemic period and 10.4% (974 deaths) during the pandemic. The risk of in-hospital death (with hospitalizations not related to COVID-19) during the pandemic was 62% higher (risk ratio [RR] 1.62; 95% CI 1.20–2.20; *p* = 0.002). At centers where the reduction in the number of “cardiological, non-COVID” admissions during the pandemic was >50% (compared to the pre-pandemic period), the risk of death was clearly higher (RR 2.74; 95% CI 2.43–3.10) than in centers where the decline was less than 50% (RR 1.21; 95% CI 1.07–1.37; *p* < 0.001) [4].

In another study on this issue, Dessie and Zewotir confirmed the importance of the coexistence of hypertension, diabetes, or other cardiovascular diseases as risk factors for death in the course of COVID-19 [5]. Hessami et al. [6] addressed more broadly the role of cardiological risk factors in the severe course of COVID-19. According to their analyses, the risk of death from COVID-19 was increased by 13 fold with the presence of acute myocardial damage (OR 13.29; 95% CI 7.35–24.03), 2.6 fold with hypertension (OR 2.60; 95% CI 2.11–3.19), almost 7 fold with heart failure (OR 6.72; 95% CI 3.34–13.52), almost 3 fold with cardiac arrhythmias (OR 2.75; 95% CI 1.43–5.25), almost 4 fold with coronary heart disease (OR 3.78; 95% CI 2.42–5.90), and 2.6 fold with other cardiovascular diseases in general (OR 2.61; 95% CI 1.89–3.62) [6]. Acute myocardial damage, cardiac arrhythmias, and the coexistence of hypertension and cardiovascular diseases, including coronary artery disease, were also risk factors for admission to the intensive care unit (ICU) with COVID-19 [6]. Liang and colleagues conducted a meta-analysis of data for patients (*n* > 22,000) with confirmed coronary artery disease [7] and found significantly increased risks for an unfavorable course with COVID-19 (OR 3.42; 95% CI 2.83–4.13; *p* < 0.001) and for mortality (OR 3.75; 95% CI 2.91–4.82; *p* < 0.001), critically severe infection (OR 3.23; 95% CI 2.19–4.77; *p* < 0.001), disease progression (OR 3.01; 95% CI 1.46–6.22; *p* = 0.003), and ICU admission (OR 2.25; 95% CI 1.34–3.79; *p* = 0.002).

An increased risk of severe COVID-19 has also been identified with dyslipidemia, obesity, or obstructive sleep apnea [8,9,10]. In one study from Estonia of influenza illness or vaccination within the 2 years before COVID-19 onset, influenza illness was a factor in a worsened prognosis, and influenza vaccination was a protective factor [10]. Among cardiac patients, several studies, including some conducted in Poland, have identified those with heart failure as being particularly at risk of severe COVID-19 infection and at high risk of both in-hospital death and death in the period after hospitalization [11,12,13,14,15,16].

Thus, the evidence strongly supports that cardiovascular diseases and risk factors for these diseases (e.g., arterial hypertension and diabetes) are associated with increased risk of severe COVID-19 and death due to COVID-19 (Figure 1).

## 4. Cardiovascular Complications of COVID-19

Cardiovascular complications because of SARS-CoV-2 infection are a separate problem from risks associated with existing cardiovascular conditions. Early analyses from the beginning of the pandemic indicated a significant impact of SARS-CoV-2 infection on damage to the cardiovascular system. In their meta-analysis of 12 studies with over 3000 patients, Zhao et al. found that the four most common cardiovascular complications of COVID-19 were myocardial damage (in 21%; 95% CI 12.3–30.0%), arrhythmias (15%; 95% CI 8.4–22.3%), features of heart failure (14%; 95% CI 5.7–23.1%), and acute coronary syndromes (1%; 95% CI 0.5–1.5%) [17]. The incidence of these complications was much higher in patients whose infection resulted in death, with signs of heart failure in 47% (95% CI 41.4–54.2%), cardiac arrhythmias in 40.3% (95% CI 1.6–78.9%), and biochemical indicators of myocardial damage in 61.7% (95% CI 46.8–76.6%) [17]. A frequent problem is the issue of myocarditis after COVID-19. Diagnosing myocarditis is difficult, but autopsy data have confirmed myocarditis in patients who have died of COVID-19, and the clinical diagnosis of myocarditis has been confirmed by MRI in up to 19% of patients with post-COVID syndrome [18].

Pulmonary embolism is another common complication observed clinically. Its characteristics in the course of COVID-19 are slightly different from the classic pulmonary manifestation of venous thromboembolism, however. Primarily, thrombotic material is found much less frequently in the deep veins of the lower limbs in patients with COVID-19. The changes are more often in the peripheral distribution of the lung, and clotting appears to arise in situ in vessels of lung areas affected by the immune–inflammatory process [19]. Suh and colleagues reported that pulmonary embolism occurred in approximately 16% of patients hospitalized for COVID-19 (95% CI 11.6–22.9%; I^2^ = 0.93), and significantly more often in the ICU (24.7%; 95% CI 18.6–32.1%) than in the regular wards (10.5%; 95% CI 5.1–20.2%) [20]. In a 2023 study in Poland, pulmonary embolism was found in over 62% (73/107) of patients admitted to the hospital for respiratory failure with COVID-19 [21]. In that study, the average proportion of lung involved in the inflammatory process was 48% (as assessed using computed tomography), but the authors found no significant correlation between the percentage of lung involvement and *n*-terminal pro-b-type natriuretic peptide or D-dimer concentration [21]. The risk of thromboembolic complications can persist after COVID-19. In a systematic review comparing the risk of thromboembolic events between patients who had had COVID-19 and those who had not, Zuin et al. [22] analyzed the data of almost 30 million patients, more than 2 million of whom had had COVID-19. With an average follow-up of 8.5 months, the cumulative incidence of pulmonary embolism among COVID-19-recovered patients was 1.2% (95% CI 0.9–1.4%; I^2^ = 99.8%), more than three times higher than in the control population (hazard ratio [HR] 3.16; 95% CI 2.63–3.79; I^2^ = 90.1%). The cumulative incidence of deep vein thrombosis among COVID-19–recovered patients was 2.3% (95% CI 1.7–3.0%; I^2^ = 99.7%), more than 2.5 fold that for the control population (HR 2.55; 95% CI 2.09–3.11; I^2^ = 92.6%) [22].

The persistence of long-term conditions after COVID-19 is currently referred to as post-COVID or long-COVID [23]. Persistent cardiovascular ailments in patients after COVID-19 include non-specific chest pain (affecting up to 21% of patients) and heart palpitations (9%). Abnormalities in echocardiography have also been described in patients with documented normal pre-infection images. The frequency of myocarditis as a COVID-19 complication is also of clinical concern, and postural orthostatic tachycardia syndrome has been described as part of long-COVID syndrome, at an undetermined frequency [23].

As these risks remain to be clarified, there is no doubt that contracting COVID-19 can lead to cardiovascular complications, either acutely fatal or with delayed, chronic effects (Figure 2).

Experts associated with the European Society of Cardiology task force for the management of COVID-19, in probing the pathophysiological mechanisms of the two-way relationship between cardiovascular diseases and COVID-19, remind us that SARS-CoV-2 binds to the angiotensin-converting enzyme (ACE) 2 receptor for entry into cells [24,25]. Organs with high ACE2 expression include blood vessels, the heart, and the lungs. Dysregulation between ACE and ACE2 activity may underlie the increased risk for patients with cardiovascular diseases of a severe course of COVID-19, as well as serve as the mechanism of some cardiovascular complications. A potential role of cathepsins and neuropilin-1, one of the most important receptors for vascular endothelial growth factor, has also been postulated. Cytokine storm, in which interleukin-6 seems to be one of the most important mediators, can also lead to a number of unfavorable changes in the cardiovascular system, causing rupture of atherosclerotic plaques with the classic consequences, for example [24,25]. Finally, acute lung injury itself may lead to increased workload on the heart, decompensating pre-existing heart failure, or provoking acute heart failure.

Thus, there can be no doubt that a patient with cardiovascular disease above all requires effective prevention of SARS-CoV-2 infection. Cases of infection require the most effective treatment possible.

## 5. Current Recommendations Regarding Vaccinations Against COVID-19

Currently, vaccination against COVID-19 is recommended for all adults aged 18 to 59 and is particularly recommended for people ≥ 60 years of age [26]. In accordance with the November 2023 Polish Ministry of Health Announcement No. 34 (referring to the recommendations of the Immunization Team of 4 September 2023; the WHO recommendation of 10 November 2023; European Centre for Disease Control and Prevention and Control (ECDC) information of 17 August 2023; and the 31 October 2023 announcement from the European Medicines Agency), defined groups are recommended to have received the updated booster vaccine for the XBB.1.5 subvariant [27,28,29,30,31]. These groups include people aged ≥ 60 years, people aged ≥ 12 years with immunodeficiencies or comorbidities that increase the risk of severe COVID-19, and healthcare workers who have direct contact with a patient or infectious material.

It should be emphasized that comorbidities “increasing the risk of severe COVID-19” include hypertension and cardiovascular diseases, primarily heart failure, coronary artery disease, cardiac arrhythmias, cardiomyopathy, and valvular defects. A cardiac patient is a person at risk of severe COVID-19 infection itself, as well as the occurrence of acute and long-term cardiovascular complications. It is also worth noting that the Ministry’s announcement concerns primarily the advisability of administering a booster dose of the vaccine, updated with the currently dominant virus variant. From the beginning of the pandemic until this position was established, many variants had already been identified, including the delta (B.1.617.2) and omicron (B.1.1.529) variants. The omicron variant turned out to be less virulent than its predecessor, causing proportionally fewer deaths and severe infections, but at the same time, it broke the immunity resulting from the baseline vaccination regimen [32,33]. With this breakthrough infection rate, the topic of booster doses arose, and the fourth dose of the Comirnaty mRNA vaccine was updated for the omicron variant, and the fourth dose of the Spikevax mRNA vaccine was a continuation of vaccinations started with a specific mRNA vaccine or as a complement to the Vaxzevria vector vaccine or the Nuvaxovid protein vaccine. The recommendation was that the initial vaccination regimens with older generation vaccines should be supplemented and updated with mRNA vaccines [34].

Since the availability of specific vaccines and the need to administer subsequent booster doses depend primarily on current Ministry of Health policy and the genetic variability of the virus, which are difficult to predict, we will focus below on the issue of the safety and effectiveness of vaccination against COVID-19 in the context of cardiac patients. The results of this analysis support the recommendation of vaccinations in general and periodic booster vaccinations in line with current guidance for our patients.

## 6. Safety and Effectiveness of Vaccinations Against COVID-19

In 2021, Cai et al. published a collective analysis of the effectiveness and safety of various COVID-19 vaccination regimens in use at that time [35]. They showed that the overall effectiveness of vaccinations was approximately 70% and that mRNA vaccines were the most effective (94.29%) [35]. The authors also commented on the issue of adverse post-vaccination reactions. This broad term includes mild symptoms or those with an intuitively predictable occurrence (e.g., pain at the injection site). Symptoms such as pain at the injection site, headache, local swelling, muscle pain, joint pain, a general feeling of malaise, chills, or fever occurred at different frequencies for individual preparations. Some of the side effects described were more common after the second or booster dose of the vaccine. In the vast majority of cases, these side effects were mild, and all were transient and self-limiting [35].

Serious side effects that pose a real threat have been observed extremely rarely. Thromboembolic complications were observed basically only after vector vaccines; in the case of the Ad26.COV2.S vaccine, their incidence was 75 per 1 million doses, and in the case of AZD1222, their rate was 21 per 1 million doses [35]. Complications in the form of myocarditis were observed after both vector vaccines and mRNA vaccines, but their incidence was even lower than thromboembolic complications—2–3 cases per 1 million doses administered [35]. Gluckman et al. compared the risk of myocarditis after mRNA vaccines with the benefits obtained from vaccination [36]. They calculated that among 1 million boys and men aged 12–19 years receiving a second dose of a vaccine based on mRNA technology, there would be 39 cases of myocarditis and at the same time 560 fewer hospitalizations, 138 fewer admissions to ICUs, and 6 fewer deaths from COVID-19. Analyses from US studies also provided interesting data, comparing the risk of cardiovascular complications in children and young adults (defined as myocarditis, pericarditis, and multisystem inflammatory syndrome) in the course of COVID-19 to risks after administration of a second dose of the mRNA vaccine. In boys aged 12–18 years, this risk was 2–6 times higher with infection than after vaccination, and among men aged 18–29 years, it was 7–8 times higher during the course of COVID-19 [37,38]. Most people with post-vaccination myocarditis recover within 90 days of diagnosis [38]. Moreover, despite some concerns about potential cerebrovascular events following vaccines developed using mRNA technology, there has been no evidence of a cause-and-effect relationship between these vaccines and the risk of stroke [39,40,41,42]. The literature from the period of the COVID-19 pandemic contains case reports and case series of thromboembolic complications after vaccination against COVID-19 [43,44,45,46,47,48,49,50,51,52,53,54,55,56,57,58,59]. However, most cases involved older patients and vector vaccines, with a mechanism of post-vaccination thrombocytopenia (i.e., vaccine-induced immune thrombotic thrombocytopenia, or VITT), an immune reaction similar to some extent to post-heparin thrombocytopenia. This complication is potentially dangerous but extremely rare, and the number of case reports is most likely due to the unprecedented scale of widespread vaccination, which yielded a large number of people for analysis [60,61,62]. Ultimately, in 2021, universal vaccination contributed to an estimated 63% reduction in global mortality related to COVID-19 and prevented 19.8 million deaths worldwide [63]. These numbers are also impressive in the estimates carried out for Poland, with an estimated 61,000 fewer deaths in 2021 thanks to vaccinations [64]. At the same time, the incidence of death among unvaccinated people compared to those who received two doses (data for 2021) was 10 times higher [64].

The impact of vaccination on long-COVID syndrome is also worth noting. Watanabe et al. analyzed the results of six observational studies involving 536,291 unvaccinated and 84,603 patients vaccinated against SARS-CoV-2 and six observational studies involving 8199 patients with long COVID syndrome [65]. Receiving two doses of the vaccine was associated with a lower risk of long-COVID syndrome compared with being unvaccinated (OR 0.64; 95% CI 0.45–0.92) or having received only one dose (OR 0.60; 95% CI 0.43–0.83) [65]. People who received the full primary series of vaccinations in case of infection had a lower risk of chronic fatigue syndrome (OR 0.62; 95% CI 0.41–0.93) and respiratory dysfunction (OR 0.50; 95% CI 0.47–0.52) [65] (Table 2). Among patients with long-COVID syndrome, 50% did not experience any beneficial changes after vaccination, but 20% experienced a reduction in symptoms [65].

There is no doubt that the implementation of global vaccinations against COVID-19 was a breakthrough in the fight against the pandemic. As with any medical intervention, vaccination also carries a certain risk of side effects. Even with the unprecedented scale of widespread vaccination, however, the occurrence of significant, life-threatening side effects was extremely rare with the COVID-19 vaccines, so there is little reason to fear these vaccinations. At the same time, acknowledging their effectiveness and that patients with cardiovascular diseases are particularly at risk if exposed to infection of a severe course, we should actively recommend vaccinations and booster doses in accordance with current recommendations from the relevant public health and medical experts. The authors of European guidelines on heart failure have taken a clear position on this issue, recommending consideration of vaccination of patients with heart failure against influenza, pneumococci, and SARS-CoV-2, and the guidelines on pulmonary hypertension also recommend similar practices [66,67].

The COVID-19 vaccines are among the vaccinations recommended by the Polish Society of Vaccinology in the 2024 schedule for adults [68]. In that document, a section devoted to patients with cardiovascular diseases recommends the COVID-19 p/c vaccine in the number of doses consistent with vaccination history and current recommendations [68].

## 7. Current Options for Pharmacotherapy of SARS-CoV-2 Infection

Vaccinations are the basis of effective prevention. Being aware of the potentially severe course of COVID-19 in a cardiac patient still calls for up-to-date knowledge on the principles of treatment of a SARS-CoV-2 infection. In November 2023, the WHO published guidelines on the principles of COVID-19 therapy [69]. The authors identified three possible courses of infection, as follows:Critical, understood as involving acute respiratory failure, shock, and need for mechanical ventilation or non-invasive ventilatory support or vasopressors;Severe, with oxygen saturation declining to <90% when breathing atmospheric oxygen, pneumonia, or signs of severe respiratory effort (one condition is enough);Non-severe, understood as not meeting any of the criteria for a severe and critical course.

Current guidelines on the principles of pharmacotherapy in the event of diagnosis of SARS-CoV-2 infection also require determination of the risk of hospitalization for COVID-19. It is assumed that the high-risk group (6%) includes patients with diagnosed immune deficiency syndromes, after organ transplantation, and with autoimmune diseases receiving immunosuppressive drugs. The intermediate risk group (3%) includes people over age 65 years and people with obesity, diabetes, active cancer, numerous comorbidities, dementia, or chronic diseases of the respiratory system, kidneys, liver, or cardiovascular system. People who do not meet the criteria for being at high or intermediate risk of hospitalization have a low risk (0.5%). They probably constitute the majority of the general population, but cardiac patients, by definition, are at intermediate cardiovascular risk, which significantly influences potential decisions regarding the possibility of pharmacotherapy of SARS-CoV-2 infection [69].

Currently, if a non-severe SARS-CoV-2 infection is detected in a patient at high risk of hospitalization, the drug of choice is nirmatrelvir/ritonavir, but the indications for its use depend on the level of risk for hospitalization due to COVID-19. The guidelines also mention remdesivir and molnupiravir, but the relevance of these drugs has decreased significantly compared to the initial period of their use [69]. Molnupiravir (oral form) and remdesivir (intravenous form) were available only through the Government Agency for Strategic Reserves during the pandemic, and in the fall of 2023, it was no longer possible to obtain these drugs. In the case of a severe or critical course of infection, corticosteroids and interleukin 6 receptor blockers (tocilizumab is registered for COVID-19 therapy) or baricitinib (a JAK inhibitor, still not officially registered in Poland for this purpose) should be used. Tocilizumab has a number of other indications, apart from COVID-19. Baricitinib is also registered for rheumatological indications, and the use of either drug is justified only in the case of a severe or critical course of COVID-19. Nirmatrelvir/ritonavir, in turn, is available in pharmacies but expensive and not reimbursed; however, it is worth attention in the context of potential use in cardiac patients with non-severe SARS-CoV-2 infection (Figure 3).

The above matrix takes into account the clinical form of the infection and the risk of hospitalization in the case of a non-severe infection. The matrix includes only drugs currently available in Poland, described in the WHO document as strategies with strong or weak/conditional recommendations justifying their use. The matrix does not include remdesivir and molnupiravir, which were unavailable in Poland at the time of creation of this document (comment in the text). The figure is based on Therapeutics and COVID-19: living guideline [69]; the source material also includes information about molecules undergoing clinical trials and about molecules for which we have weak/conditional or strong evidence of lack of justification for use. The figure was created with BioRender.com.

## 8. Nirmatrelvir/Ritonavir

Nirmatrelvir/ritonavir is a combination of two molecules. Nirmatrelvir is an inhibitor of the SARS-CoV-2-3CL protease and thus inhibits viral replication at the proteolysis stage (i.e., before viral RNA replication). Ritonavir is a well-known antiviral drug originally developed to treat HIV infection and is also a strong CYP3A4 inhibitor, which increases the concentration of nirmatrelvir by slowing down its metabolism [69,70]. The key study for the current role of this combination in COVID-19 therapy was EPIC-HR, which showed that use of nirmatrelvir/ritonavir within 3–5 days of the onset of symptoms effectively reduces the risk of hospitalization and death [71]. Subsequent studies confirmed the effectiveness of the drug mainly in groups at high risk of severe infection. For example, Najjar-Debbiny et al. conducted a study in Israel of more than 180,000 patients, 79% of whom were aged ≥ 60 years and 75% of whom were vaccinated. Of the cohort, 4737 (2.6%) people were treated with nirmatrelvir/ritonavir. Both the treatment and optimal COVID-19 vaccination status were associated with a significant reduction in the risk of severe COVID-19 or death, with modified HRs of 0.54 (95% CI 0.39–0.75) for the drug combination and 0.20 (95% CI 0.17–0.22) for vaccination [72]. Another study confirmed the effectiveness of the drug in the era of the omicron variant, in various age groups and regardless of vaccination status [73,74].

Nirmatrelvir/ritonavir has also been shown to be more effective than molnupiravir [75]. Overall treatment tolerability in studies has been good, and the duration of therapy is short (5 days of twice-daily use). However, the drug’s strong impact on cytochrome p450 CYP 3A4 activity and potential drug interactions require additional comment. Before initiating nirmatrelvir/ritonavir treatment, it is necessary to assess whether other drugs the patient uses pose the potential for a significant pharmacokinetic interaction. For this purpose, as recommended by WHO guidelines, a special tool is available online, the Liverpool COVID-19 drug interaction checker (https://www.covid19-druginteractions.org/checker (accessed on 20 May 2025)) [76]. Important interactions that should always be kept in mind include those with oral anticoagulants. In the case of warfarin, unpredictable fluctuations in international normalized ratio values may occur, necessitating increased frequency of index monitoring; most often, the warfarin dose should be reduced or bridging therapy with low-molecular-weight heparin should be considered (or alternatively, other antiviral drugs, depending on their availability). In the case of direct oral anticoagulants, if temporary interruption of therapy is not possible, bridging therapy with low-molecular-weight heparin is suggested [77]. Concomitant use of rivaroxaban and nirmatrelvir/ritonavir is contraindicated; in the case of apixaban and dabigatran, the risk of interaction is described as “potential”, but caution should be exercised [77,78,79].

The list of drugs that are substrates for CYP3A4 is long and includes statins, calcium channel antagonists, clopidogrel, and the already mentioned anticoagulant drugs (Table 3). Current WHO guidelines recommend the use of nirmatrelvir/ritonavir in patients with a non-severe course of SARS-CoV-2 infection who are at high risk of hospitalization. In patients at intermediate risk of hospitalization, the use of this drug should be considered.

This latter group will most often include cardiac patients. In the group at high risk of hospitalization, the use of nirmatrelvir/ritonavir reduces the risk of hospitalization and death, but for the intermediate-risk group, we have only strong evidence that a reduced need for COVID-19 hospitalization is possible [71]. The latest analyses also indicate that this drug is not effective as part of post-exposure therapy in patients who are exposed to the disease but do not yet present symptoms of infection [80]. Adverse effects of nirmatrelvir + ritonavir are described as rare and transient, mainly concerning the gastrointestinal tract (taste disorders, diarrhea, and vomiting) or dizziness. However, this drug is still new and used infrequently, and it is therefore reasonable to emphasize the importance of pharmacovigilance. Nevertheless, taking into account the necessary caution related to potential drug interactions, nirmatrelvir/ritonavir remains the most effective treatment in the early phase of SARS-CoV-2 infection, and its use should be considered in cardiac patients, regardless of vaccination status.

In order to optimize the safety of drug use, we propose the following practical approach to the problem of deciding on the implementation of treatment:In the case of COVID-19 diagnosis in a patient with symptoms of infection, assess the group at increased risk of severe disease (Figure 3)If the patient qualifies for nirmatrelvir/ritonavir, analyze the chronic pharmacotherapy regimen in the context of potentially significant drug interactions (Table 3).Drug coadministration:
In the absence of a risk of interaction, propose the use of nirmatrelvir/ritonavir regardless of vaccination status.In the case of a significant risk of interaction, consider the possibility of temporary modification of the pharmacotherapy regimen (Table 4).

## 9. Limitations

Our narrative review has certain limitations. First of all, the text is not a systematic review or meta-analysis. It is the opinion of multidisciplinary experts whose primary goal is to emphasize the importance of COVID-19 prevention in the population of patients with cardiovascular diseases. It can be assumed that the studies we refer to are heterogeneous. After all, they refer to different periods of the pandemic. We do not analyze individual subpopulations or local conditions resulting from the construction of health care systems separately, although the starting point for our considerations is the situation in Poland. We do not comment on specific, current recommendations for booster doses. These recommendations are dynamic, and the availability of individual vaccines also varies. Moreover, when analyzing the impact of vaccination and treatment on the onset of long COVID syndrome, it should be remembered that its pathophysiology remains a subject of research. Finally, there are still some gaps in the evidence in the publications cited by us, and difficulties in establishing cause–effect relationships between post-infection and post-vaccination events can be raised. However, the primary goal of our article is to draw cardiologists’ attention to the relationship between COVID-19 and the occurrence of cardiovascular complications, the justification for promoting vaccinations, and the possibility of developing targeted treatment in high-risk groups. Taking all of this into account, we believe that the conclusions presented below are fully justified given our current state of knowledge.

## 10. Conclusions

The COVID-19 pandemic has formally ended, but SARS-CoV-2 remains in circulation and is undergoing further mutations. New variants of the virus currently seem less virulent, but their development emphasizes the need to update vaccines and periodically administer new booster doses. COVID-19 is slowly becoming similar to influenza, and SARS-CoV-2 itself will likely become a pathogen known for causing periodic epidemics. Taking into account the strong connection between the risk of SARS-CoV-2 infection and existing cardiovascular problems and the potential complications of infection related to the circulatory system, as proven during the pandemic, we believe that cardiologists should recommend booster doses of COVID-19 vaccines for their patients. The accelerated development of COVID-19 vaccines was a global imperative, and regulatory and medical decisions were based on calculating benefits and risks [86]. While regulatory standards for safety and efficacy evaluations were maintained during clinical trials, mass vaccination campaigns revealed rare side effects that were not detectable in pre-authorization studies due to their low incidence [87,88,89,90,91]. For example, myocarditis after mRNA vaccines was mainly observed in younger males, while adenoviral vector vaccines were associated with rare cases of thrombosis with thrombocytopenia. Although both side effects are classified as very rare, these findings highlight the importance of ongoing pharmacovigilance, especially in high-risk groups. NVX-CoV2373 (Novavax) stands out for its favorable safety and efficacy profile among vaccines available in Europe and the US. Phase 3 trials demonstrated 89.7% efficacy against symptomatic COVID-19, with most mild and transient adverse events. While mRNA vaccines have acceptable risk profiles, the recombinant subunit vaccine (Novavax) is a strong candidate for high-risk populations, combining high efficacy with safety [86]. The situation is dynamic, as recommendations regarding the type of vaccines used may change each season; therefore, it is probably best to respond in general terms and refer to current national and international recommendations. Moreover, a cardiac patient, by definition, is at least indirectly at risk of hospitalization in the course of a SARS-CoV-2 infection, even if the initial COVID-19 course is mild. According to current WHO guidelines, the use of nirmatrelvir/ritonavir should be considered in this group of patients, accounting for potential interactions with other concomitantly used drugs. This therapy has the potential to reduce the risk of hospitalization and, in the case of high-risk patients, to reduce mortality, regardless of vaccination status (Figure 4). It is worth emphasizing that, as with all new developments in medicine, further observation, research, and pharmacovigilance remain very important.

## Figures and Tables

**Figure 1 vaccines-13-00554-f001:**
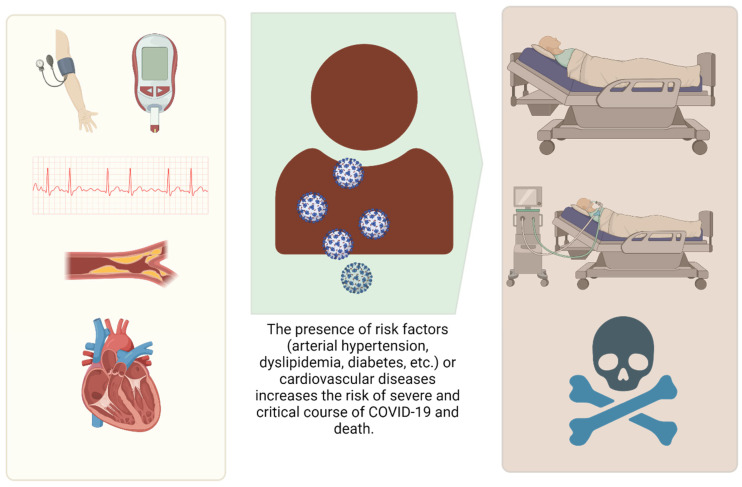
The impact of classic risk factors and cardiovascular diseases on the prognosis of patients with SARS-CoV-2 infection. Figure created with BioRender.com.

**Figure 2 vaccines-13-00554-f002:**
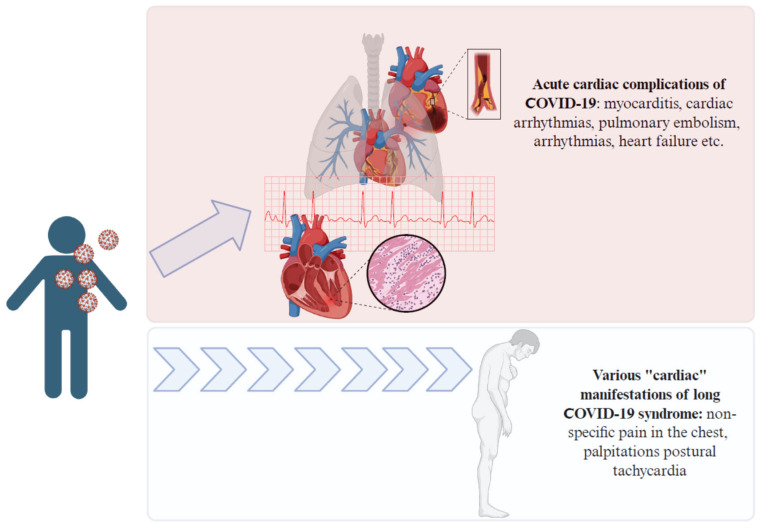
Acute and chronic complications of COVID-19. Figure created with BioRender.com.

**Figure 3 vaccines-13-00554-f003:**
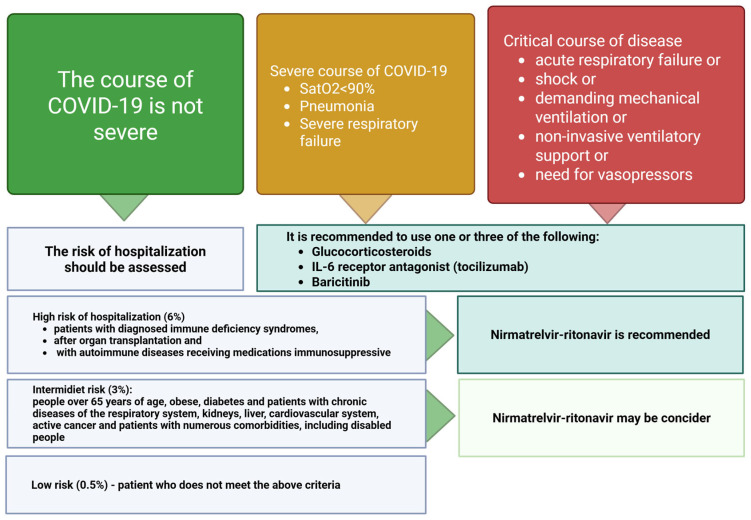
Pharmacotherapy matrix for COVID-19 patients. Figure created with BioRender.com.

**Figure 4 vaccines-13-00554-f004:**
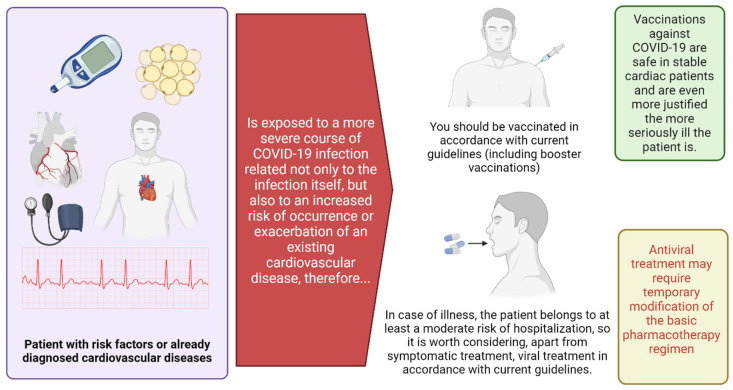
Central figure. A patient with cardiovascular diseases is particularly vulnerable to severe COVID-19. This fact is an argument that clearly supports the validity of preventive vaccinations. If an infection is diagnosed, antiviral treatment should be considered. Vaccinations and drug therapy should be administered according to current recommendations. Figure created with BioRender.com.

**Table 1 vaccines-13-00554-t001:** The impact of hypertension, diabetes, and cardiovascular diseases on the course of COVID-19 infection in individual age groups. Based on the analysis by Bae S. et al. [3].

Comorbidities	Age	Severe Course of COVID-19	Death from COVID-19	Death from or Severe Course of COVID-19
		OR (95% CI)		
**Arterial hypertension**	<50 r. ż	3.14 (2.16–4.55)	6.13 (4.01–9.39)	3.49 (2.49–4.88)
	50–60 r. ż	2.43 (1.72–3.44)	2.81 (1.97–4.01)	2.61 (2.03–3.37)
	≥60 r. ż.	1.54 (1.35–1.76)	2.10 (1.67–2.64)	1.86 (1.55–2.23)
	all	2.42 (1.98–2.96)	2.60 (2.11–3.20)	2.50 (2.15–2.90)
**Diabetes**	<50 r. ż	3.24 (1.96–5.36)	5.31 (3.22–8.26)	3.49 (2.24–5.45)
	50–60 r. ż	2.73 (1.59–4.7)	2.22 (1.82–2.72)	2.37 (1.83–3.08)
	≥60 r. ż.	1.44 (1.11–1.86)	1.76 (1.27–2.44)	1.65 (1.34–2.03)
	all	2.47 (1.86–3.27)	2.11 (1.63–2.73)	2.25 (1.89–2.69)
**CVD**	<50 r. ż	4.87 (3.31–7.16)	7.80 (4.06–15.00)	5.66 (4.12–7.79)
	50–60 r. ż	3.09 (2.30–4.17)	3.14 (1.12–8.78)	3.30 (2.16–5.05)
	≥60 r. ż.	1.73 (1.01–2.90)	2.46 (1.91–3.17)	2.10 (1.68–2.61)
	all	3.15 (2.34–4.25)	3.23 (2.26–4.57)	3.11 (2.55–3.79)

CVD—cardiovascular diseases, defined as “history of cardiovascular diseases or its synonyms, such as coronary heart disease” (based on the description of the meta-analysis methodology). OR, odds ratio. CI, confidence interval.

**Table 2 vaccines-13-00554-t002:** The impact of COVID-19 vaccination on the occurrence of long-term complications of the infection (so-called long-COVID in general and its individual manifestations).

Analyzed Effect	2 Doses of the Vaccine vs. Unvaccinated OR (95% CI)	2 vs. 1 Dose of the Vaccine OR (95% CI)	1 Dose of the Vaccine vs. Unvaccinated OR (95% CI)
**Long-COVID syndrome**	0.64 (0.45–0.92)	0.60 (0.43–0.83)	0.90 (0.80–1.01)
**Chronic fatigue**	0.62 (0.41–0.93)		
**Respiratory disorders**	0.50 (0.47–0.52)		
**Cardiovascular disorders**	0.59 (0.23–1.49)		
**Gastrointestinal disorders**	0.62 (0.35–1.10)		
**Metabolic disorders**	0.58 (0.24–1.41)		
**Musculoskeletal disorders**	0.63 (0.26–1.52)		
**Neurological disorders**	0.61(0.31–1.20)		
**Cognitive disorders**	0.49 (0.15–1.62)		

OR, odds ratios. CI, confidence interval.

**Table 3 vaccines-13-00554-t003:** Potential interactions between selected drugs used in COVID-19 therapy and selected cardiological drugs. Based on the Liverpool COVID-19 drug interaction checker (https://www.covid19-druginteractions.org/checker (accessed on 4 September 2024) [76].

	Baricitinib	Nirmatrelvir/Ritonavir (5 Days)	Remdesivir	Tocilizumab
*Acenocoumarol*				
*Amiodarone*		Possible increase in amiodarone concentration		
*Amlodipin*				
*Apixaban*				
*Acetylsalicylic acid*				
*Atorvastatin*				
*Candesartan*				
*Clopidogrel*		Possible increase in active metabolite of clopidogrel concentration		
*Dabigatran*				
*Dalteparin*				
*Digoxin*				
*Doxazosin*				
*Empagliflozin*				
*Enalapril*				
*Enoxaparin*				
*Eplerenone*		Possible increase in eplerenone concentration		
*Ezetimibe*				
*Furosemide*				
*Ivabradine*		Possible increase in ivabradine concentration		
*Labetalol*				
*Lisinopril*				
*Losartan*				
*Lovastatin*		Possible increase in lovastatin concentration		
*Metoprolol*				
*Nebivolol*				
*Omega-3*				
*Prasugrel*				
*Propafenone*		Possible increase in simvastatin concentration		
*Ramipril*				
*Rivaroxaban*		Increase in rivaroxaban concentration		
*Rosuvastatin*				
*Simvastatin*		Possible increase in simvastatin concentration		
*Spironolactone*				
*Tadalafil*		Possible increase in tadalafil concentration		
*Telmisartan*				
*Ticagrelor*		Possible increase in ticagrelor concentration		
*Torasemide*				
*Valsartan*				
*Varfarin*				

The green color means no significant interaction, the yellow color means potential weak interaction, the orange color means potential interaction (be careful!), and the red color means significant interaction—do not combine the drugs. The red fields additionally describe the direction of changes in the concentration of the cardiac drug. It should be emphasized that the quality of evidence for a given interaction is very low, the effects of co-administration have not been clinically tested, and the description of the interaction is a consequence of the analysis of the pharmacokinetic properties of the molecules.

**Table 4 vaccines-13-00554-t004:** Suggestions for temporary modifications of the drug treatment regimen in patients with cardiovascular diseases in the case of indications for the use of nirmatrelvir/ritonavir. The choice of the described molecules is made subjectively by the authors. The proposed strategies of action were developed based on the analysis of selected publications [81,82,83,84,85].

Cardiovascular Drug	Concerns About Possible Interaction	Proposed Strategy
Amiodarone	There is a risk of serious toxicity AND stopping the drug does not mitigate the interaction due to amiodarone’s prolonged half-life	Do not use nirmatrelvir/ritonavir (consider a different antiviral drug, if possible and available)
Other antiarrhythmic drugs as propafenone, quinidine, and dronedarone	There is a risk of serious toxicity	Administrate nirmatrelvir/ritonavir ONLY if the other drug can be safely paused or replaced. Resume the other drug 3–5 days after stopping nirmatrelvir/ritonavir
Rivaroxaban, apixaban, ticagrelor, and clopidogrel	There is a risk of serious toxicity	Administrate nirmatrelvir/ritonavir ONLY if the other drug can be safely paused or replaced; resumption is possible 3–5 days after stopping nirmatrelvir/ritonavir. Acetylsalicylic acid and prasugrel remain safe antiplatelet drugs; low-molecular-weight heparins remain safe anticoagulant drugs
Warfarin (acenocumarole) and dabigatran	Toxicity is possible	If possible, before administering nirmatrelvir/ritonavir, stop and replace the other drug (low-molecular-weight heparins remain a safe alternative). Dose reduction (dabigatran) or close monitoring (VKA) remain acceptable management strategies. Resumption of cardiovascular drugs is possible 3–5 days after stopping nirmatrelvir/ritonavir
Lercanidipine	There is a risk of serious toxicity	Administer nirmatrelvir/ritonavir ONLY if the other drug can be safely paused or replaced; resumption is possible 3–5 days after stopping nirmatrelvir/ritonavir
Other calcium channel blockers than lercanidipine	Toxicity is possible	Monitoring blood pressure values (discontinuing the drug in case of excessive drop in blood pressure) seems to be an acceptable treatment strategy
Eplerenone	There is a risk of serious toxicity	Administer nirmatrelvir/ritonavir ONLY if the other drug can be safely paused or replaced; resumption is possible 3–5 days after stopping nirmatrelvir/ritonavir. Spironolactone remains a safe alternative
Lovastatin and simvastatin	There is a risk of serious toxicity	Administer nirmatrelvir/ritonavir ONLY if the other drug can be safely paused or replaced; resumption is possible 3–5 days after stopping nirmatrelvir/ritonavir. Pravastatin, fluvastatin, and pitavastatin remain safe alternatives. Ezetimibe also seems to remain safe, as does fenofibrate
Atorvastatin and rosuvastatin	Toxicity is possible	If possible, before administering nirmatrelvir/ritonavir, stop and replace the other drug (see above). Resumption is possible 3–5 days after stopping nirmatrelvir/ritonavir
Ivabradine, Ranolazine, Tadalafil, and Sildenafil (pulmonary hypertension)	There is a risk of serious toxicity	Administer nirmatrelvir/ritonavir ONLY if the other drug can be safely paused or replaced; resumption is possible 3–5 days after stopping nirmatrelvir/ritonavir. In case of ivabradine and ranolazine, temporarily discontinuing the drug seems to be a reasonable course of action. In the context of pulmonary hypertension therapy, consultation with a reference center seems to be warranted
Sacubitryl/valsartan, indapamid, and doxazosine	Toxicity is possible	Monitoring blood pressure values (discontinuing the drug in case of excessive drop in blood pressure) seems to be an acceptable treatment strategy
Rifampicine (infectious endocarditis)	There is a risk of serious toxicity AND stopping the drug does not mitigate the interaction due to prolonged half-life	Do not use nirmatrelvir/ritonavir (consider a different antiviral drug, if possible and available)
Digoxin	Toxicity is possible	If possible, before administering nirmatrelvir/ritonavir, stop and replace the other drug. Resumption is possible 3–5 days after stopping nirmatrelvir/ritonavir

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
