# Peer review of "The Impact of COVID-19 and the Practical Importance of Vaccinations and Nirmatrelvir/Ritonavir for Patients with Cardiovascular Disease"

_vaccines, 2025, doi:10.3390/vaccines13060554_

Round 1
Reviewer 1 Report
Comments and Suggestions for Authors
This review put forward reference discussions and clinical practice suggestions regarding high risk population- namely people with cardiovascular disease, older age with or without metabolic syndrome – they recommended vaccination, clinical monitoring and anti-viral drug treatment for the COVID-19 exposure. Table 3 is a useful compilation of possible different drug interferences, figure 4 is a good visual streamline of clinical practice.
However, due to the urgent nature of the COVID-19 pandemic, different COVID-19 vaccines are not always thoroughly tested before being administered to the general public, so that the side-effects of some of the COVID-19 vaccines are just being published recently. Please make this a clear point in your discussion and name a few effective and low toxicity vaccines to be included in your guideline.
Author Response
Dear Reviewer
Thank you very much for your review of our article. Below we address your comments and describe the modifications we have made.
Comments 1: “This review put forward reference discussions and clinical practice suggestions regarding high risk population- namely people with cardiovascular disease, older age with or without metabolic syndrome – they recommended vaccination, clinical monitoring and anti-viral drug treatment for the COVID-19 exposure”
Response 1: We are very pleased that the structure of our article leaves no doubt as to its primary purpose. We want to draw the attention of doctors treating patients with cardiovascular and metabolic diseases to the need to update their knowledge about the possibility of vaccination against COVID-19, as well as available treatment options.
Comments 2: Table 3 is a useful compilation of possible different drug interferences, figure 4 is a good visual streamline of clinical practice.
Response 2: Thank you for your words of appreciation. Developing a table of potential drug interactions was a challenge, but we believe that collecting such data in one place can be useful. We also thank you for your words of appreciation regarding our original graphics.
Comments 3: However, due to the urgent nature of the COVID-19 pandemic, different COVID-19 vaccines are not always thoroughly tested before being administered to the general public, so that the side-effects of some of the COVID-19 vaccines are just being published recently. Please make this a clear point in your discussion and name a few effective and low toxicity vaccines to be included in your guideline.
Response 3: Thank you for these words of constructive criticism. It is difficult to talk about adverse effects of vaccinations in today's world without feeding groups that create black PR. Of course, adverse effects do occur, including significant ones. However, the cost-benefit balance speaks in favor of vaccination, especially in groups at increased risk of severe disease. In accordance with your suggestion, we have expanded this thread. We have added additional explanations along with several references to the conclusion (lines 448-461). We hope that the text additions we have proposed will be considered appropriate.
We have also made a few additions in accordance with the editors' wishes, but they do not affect the substantive tone of the whole.
We hope that after all the corrections have been made, our article will be considered worthy of publication.
Best regards,
Authors
Reviewer 2 Report
Comments and Suggestions for Authors
Summary
The manuscript discusses the impact of COVID-19 on patients with cardiovascular disease and highlights the importance of vaccination and antiviral treatment, particularly nirmatrelvir/ritonavir. It reviews empirical observations and clinical trial results showing that cardiovascular disease patients are at higher risk for severe COVID-19 outcomes. The paper argues that vaccinations help mitigate these risks and emphasizes updated public health recommendations. It also examines treatment options, particularly antiviral therapies, and discusses drug interactions that must be considered in cardiac patients. The manuscript concludes that cardiologists should actively promote COVID-19 vaccinations for their patients and consider antiviral treatments when infection occurs.
Comments:
- The manuscript presents a timely and clinically relevant discussion; however, it does not introduce groundbreaking insights. It primarily synthesizes existing research. Strengthening the discussion with new data or perspectives could enhance its impact.
- While the manuscript effectively discusses both aspects, the discussion of treatment strategies could be expanded with more comparison among available antiviral agents.
- The paper highlights the importance of monitoring drug interactions with nirmatrelvir/ritonavir, but additional guidance or a decision-making algorithm for clinicians would strengthen its practical utility.
- Incorporate recent statistics on COVID-19 vaccination and antiviral effectiveness. If possible, reference studies from 2024 or 2025 to ensure the manuscript remains up-to-date.
- Expand the discussion on antiviral treatments beyond nirmatrelvir/ritonavir. Including comparisons with molnupiravir and remdesivir (even if they are no longer widely used) would give a more holistic view.
- Some tables (e.g., those referencing cardiovascular disease risk) could benefit from better formatting. Consider using clearer column headers and ensuring numerical data is spaced properly for readability.
- The manuscript makes a strong case for vaccination and antiviral treatments, but the conclusion could explicitly reaffirm the necessity for updated booster doses and vigilance in treating cardiac patients with COVID-19.
- Consider including a brief recommendation for further research or clinical guidelines, which would reinforce the paper’s importance.
Author Response
Dear Reviewer
Thank you very much for your review of our article. Below we address your comments and describe the modifications we have made.
Comments 1: The manuscript presents a timely and clinically relevant discussion; however, it does not introduce groundbreaking insights. It primarily synthesizes existing research. Strengthening the discussion with new data or perspectives could enhance its impact.
Response 1: Of course, our narrative review is not groundbreaking. In fact, we see the study we have prepared as more of a groundwork. Unfortunately, awareness of the validity of vaccination - not only against COVID-19, but also against other pathogens such as influenza - still leaves much to be desired in Poland. The article you are reviewing is one of a series on the validity of vaccination in preventing not only infections, but also exacerbations of chronic cardiovascular diseases. In Polish Journal of Cardiology, we have already published a document on the validity of vaccinating patients with cardiovascular diseases against penumococcus (Mamcarz, A., WeÅ‚nicki, M., Drożdż, J. et al. The role of pneumococcal vaccination in reducing cardiovascular risk in cardiac patients: Expert opinion of the Prevention Committee of the Polish Cardiac Society supported by the Polish Vaccinology Society. Kardiologia polska, 81(10), 1038–1046. https://doi.org/10.33963/v.kp.96718). The document on vaccination against influenza is also under review. We want cardiologists to look at the problem of prevention more broadly than before.
Comments 2: While the manuscript effectively discusses both aspects, the discussion of treatment strategies could be expanded with more comparison among available antiviral agents.
Response 2: Real data on the comparison of individual antiviral treatment strategies are still very scarce. To our knowledge, nirmatrelvir/ritonavir is currently the most widely studied therapeutic option. Molnupinavir and remdesivir are currently unavailable in Poland, so our experience with this drug is very limited. We therefore limit ourselves to data that we recognize as verified and confirmed. Moreover, the basic topic of our study is the prevention of infection in general (hence the focus on vaccinations) and of severe infection in the high-risk population (hence the focus on nirmatrelvir/ritonavir). We hope you find these explanations sufficient.
Comments 3: The paper highlights the importance of monitoring drug interactions with nirmatrelvir/ritonavir, but additional guidance or a decision-making algorithm for clinicians would strengthen its practical utility.
Response 2: It is hard for us to disagree with this remark. In practice, decisions regarding the inclusion of the drug in question are always individual, in Poland the main limiting factor is the very high price of the drug. However, following your suggestion, we decided to propose a basic decision-making algorithm.
“In order to optimize the safety of drug use, we propose the following practical approach to the problem of deciding on the implementation of treatment:
1) In the case of COVID-19 diagnosis in a patient with symptoms of infection, assess the group at increased risk of severe disease (Fig. 3)
2) If the patient qualifies for nirmatrelvir/ritonavir, analyze the chronic pharmacotherapy regimen in the context of potentially significant drug interactions (Table 3).
3) Drug prescription:
- a) in the absence of a risk of interaction, propose the use of nirmatrelvir/ritonavir regardless of vaccination status.
- b) in the case of a significant risk of interaction, consider the possibility of temporary modification of the pharmacotherapy regimen”
Comments 4: Incorporate recent statistics on COVID-19 vaccination and antiviral effectiveness. If possible, reference studies from 2024 or 2025 to ensure the manuscript remains up-to-date.
Response 4: To the best of our knowledge, we refer to current data.
Comments 5: Expand the discussion on antiviral treatments beyond nirmatrelvir/ritonavir. Including comparisons with molnupiravir and remdesivir (even if they are no longer widely used) would give a more holistic view.
Response 5: In line with the arguments cited above, we want to focus on the currently available options for preventing COVID-19 in general (vaccinations) and severe disease in a specific population. We mention other drugs, but they are not the main topic of this paper. We do not want to expand on the topic of unavailable drugs.
Comments 6: Some tables (e.g., those referencing cardiovascular disease risk) could benefit from better formatting. Consider using clearer column headers and ensuring numerical data is spaced properly for readability.
Response 6: We appreciate this practical tip. We have modified the form of all three tables, we hope they are now more readable
Comments 7 and 8: The manuscript makes a strong case for vaccination and antiviral treatments, but the conclusion could explicitly reaffirm the necessity for updated booster doses and vigilance in treating cardiac patients with COVID-19. Consider including a brief recommendation for further research or clinical guidelines, which would reinforce the paper’s importance.
Response 7 and 8: we have included these comments in the modified version of the last paragraph of the text (conclusions)
Once again, thank you for your in-depth analysis of our paper. We appreciate your constructive criticism and hope that the above clarifications and corrections will be considered sufficient.
Best regards,
Authors
Reviewer 3 Report
Comments and Suggestions for Authors
1) The Authors performed a review article entitled "The impact of COVID-19 and the practical importance of vaccinations and nirmatrelvir/ritonavir for patients with cardiovascular disease". The publication has therapeutic interest and contains 4 figures, 3 tables, and 80 bibliographic references.
2) Lines 36-51: Please state the objective of the article clearly, as well as the methodology used.
3) Table 3: Please do not use the term "Aspirin", that is, the international common name of the active substance, i.e., acetylsalicylic acid, should be used.
4) For the nirmatrelvir + ritonavir association, it is necessary to better characterize the profile of undesirable effects (common, infrequent and rare). Additionally, it should be noted that it is a medicinal product under additional monitoring.
Author Response
Dear Reviewer
Thank you very much for your review of our article. This is also a response 1 for a general comment 1.
Below we address your other comments and describe the modifications we have made.
Comments 2: “Lines 36-51: Please state the objective of the article clearly, as well as the methodology used.”
Response 2: Our article is an narrative review. We included some additional information: “In an era of compartmentalized medical specialties, the COVID-19 pandemic is a re-minder and underlines the validity of looking at a patient's health holistically. The aim of this narrative review is to highlight effective prevention and treatment options for COVID-19 in patients with cardiovascular conditions. In an interdisciplinary group of representatives from various specialties: internal medicine, cardiology, infectious dis-eases, virology, clinical pharmacology, we have developed a summary of current data on the impact ofCOVID-19 in patients with cardiovascular disease, as well as the efficacy and safety of vaccination and current pharmacotherapy options. The aim of this narrative review is to provide arguments supported by scientific evidence to justify the pro-motion of COVID-19 vaccination by cardiologists as well as to critically present the possibility of pharmacotherapy in the event of infection”. Hope you will find it sufficient.
Comments 3: “Table 3: Please do not use the term "Aspirin", that is, the international common name of the active substance, i.e., acetylsalicylic acid, should be used”
Response 3: We fully agree and thank you for your vigilance. The name of the drug has been corrected.
Comments 4: For the nirmatrelvir + ritonavir association, it is necessary to better characterize the profile of undesirable effects (common, infrequent and rare). Additionally, it should be noted that it is a medicinal product under additional monitoring.
Response 4: Thank you for that important remark. We have added some context : “Adverse effects of nirmatrelvir + ritonavir are described as rare and transient, mainly concerning the gastrointestinal tract (taste disorders, diarrhea, vomiting) or dizziness. However, this drug is still new and used infrequently and it is therefore reasonable to emphasize the importance of pharmacovigilance„ . Hope you will find it sufficient.
We have also made a few additions in accordance with the editors' wishes, but they do not affect the substantive tone of the whole.
We hope that after all the corrections have been made, our article will be considered worthy of publication.
Best regards,
Authors
Reviewer 4 Report
Comments and Suggestions for Authors
The research paper delivers an analysis of how COVID-19 affects cardiovascular disease patients by demonstrating higher disease severity and increased death probabilities in this particular population. The paper emphasizes the importance of SARS-CoV-2 vaccines together with nirmatrelvir/ritonavir treatment according to both current international treatment guidelines and new clinical information. The paper delivers an extensive review of COVID-19 cardiovascular complications while analyzing vaccine safety and efficacy among cardiovascular patients and discussing antiviral drug therapy along with their drug interaction profiles.
The paper demonstrates logical progression through its epidemiological findings followed by pathophysiological explanations and preventive methods and therapeutic options. The manuscript becomes more understandable through its inclusion of recent references as well as tables and figures that help position the research within modern clinical literature. The study focuses on Poland but its clinical applications remain worldwide because it relies on worldwide data and clinical standards.
Major Observations
The manuscript integrates epidemiological research with meta-analyses together with major pandemic data from the highest pandemic period. The paper does not clarify how authors selected studies for inclusion when extracting mortality rates and adverse event frequencies specifically myocarditis following vaccinations.
The analysis of cited studies for recommendations and conclusions should incorporate an assessment of heterogeneity that accounts for population differences and viral variants and healthcare environment variations which affect external validity. At least, adress this in a table or short paragraph.
The identification of long COVID along with prolonged cardiovascular complications represents a notable achievement. The analysis of vaccination-related symptom prevalence needs additional clarification regarding the ongoing uncertainty and developmental changes in long COVID definitions.
Further research should analyze whether sequelae develop immediately after illness or persist chronically and establish specific mechanisms behind these complications such as microvascular damage and autoimmune responses.
The discussion about nirmatrelvir/ritonavir presents a complete overview of the drug interactions it has with standard cardiovascular medications including statins and anticoagulants and antiarrhythmics. A tabular summary should be added to present common drug interactions together with their management strategies to assist medical practitioners.
The authors support antiviral treatment yet need to expand their recommendations with specific guidance about CVD population contraindications and references to current or past clinical trials focused on emerging viral strains.
The safety assessment of vaccines includes a detailed enumeration of adverse reactions especially the thromboembolic and myocarditis adverse events. The risk–benefit evaluation for particular subgroups including young males and elderly patients with multimorbid conditions should include exact absolute risk numbers (if available).
A discussion of public health communication strategies to address vaccine hesitancy among patients with cardiovascular comorbidities would enhance the review.
The manuscript emphasizes its focus on Polish data yet needs to enhance its discussion by clearly addressing differences in healthcare access and SARS-CoV-2 variant distribution between regions.
A section should explicitly detail the review's limitations by addressing data gaps together with publication bias and difficulties in establishing cause-effect relationships between post-infection and post-vaccination events.
Minor Observations
The document contains minor typographical errors and inconsistent punctuation usage mainly affecting tabular data by using commas instead of periods for numeric values.
Long sentences appear in the document which could become easier to read by making them shorter through syntactic simplification.
The paper references Figures 1 and 2 correctly yet their legends require clearer explanations which should define abbreviations and clinical terms for better understanding.
The meta-analytic results and vaccine safety data tables provide essential value to the study yet their interpretation could become more straightforward with additional explanatory footnotes.
The reference list covers a wide range of sources and maintains currency. The accessibility of URLs is limited because some links do not provide complete access which may prevent peer verification at this time. Some references would benefit from DOIs instead of URLs when available.
More specific clinical recommendations especially for booster vaccinations and antiviral therapy timing would benefit from alignment with the latest international consensus statements as well as a brief discussion of differences among major societies (ESC, AHA, WHO, etc.).
Author Response
Dear Reviewer
First of all, thank you for your in-depth analysis of our text. We are very pleased that its quality was generally rated high by you. Thank you for your words of constructive criticism. Below we respond to your comments.
Major Observations
Comments 1: The manuscript integrates epidemiological research with meta-analyses together with major pandemic data from the highest pandemic period. The paper does not clarify how authors selected studies for inclusion when extracting mortality rates and adverse event frequencies specifically myocarditis following vaccinations.
Response 1: The document we have prepared is a narrative review. Indeed, we did not indicate this in the original version of the text. This is not a systematic review. We have supplemented the first chapter of the text (Background) with this information.
Comments 2: The analysis of cited studies for recommendations and conclusions should incorporate an assessment of heterogeneity that accounts for population differences and viral variants and healthcare environment variations which affect external validity. At least, address this in a table or short paragraph.
Response 2: We decided to add one more paragraph: Limitations, in witch In which we describe the limitations resulting from the construction of our document. Hope you will find it sufficient.
Comments 3: The identification of long COVID along with prolonged cardiovascular complications represents a notable achievement. The analysis of vaccination-related symptom prevalence needs additional clarification regarding the ongoing uncertainty and developmental changes in long COVID definitions.
Further research should analyze whether sequelae develop immediately after illness or persist chronically and establish specific mechanisms behind these complications such as microvascular damage and autoimmune responses.
Response 3: We fully agree, but the analysis of the pathophysiological mechanism of long-COVID is not the topic of this study. However, we have also included this note in the paragraph on limitations.
Comments 4: The discussion about nirmatrelvir/ritonavir presents a complete overview of the drug interactions it has with standard cardiovascular medications including statins and anticoagulants and antiarrhythmics. A tabular summary should be added to present common drug interactions together with their management strategies to assist medical practitioners.
Response 4: This is a very valuable and valid comment. Thank you for pointing it out, we have supplemented the text with an additional table (table 4) and additional references.
Comments 5: The authors support antiviral treatment yet need to expand their recommendations with specific guidance about CVD population contraindications and references to current or past clinical trials focused on emerging viral strains.
Response 5: The primary goal of our study is to draw cardiologists' attention to the relationship between COVID-19 and the occurrence of cardiovascular complications, the justification for promoting vaccinations and the possibility of reaching for targeted treatment in the high-risk group. Detailed guidelines for vaccinations and booster doses or updates on COVID-19 pharmacotherapy are discussed in other documents. One such document is the cited publication [86]: "Management of SARS-CoV-2 Infection-Clinical Practice Guidelines of the Polish Association of Epidemiologists and Infectiologists, for 2025". We are expanding on the topic of nirmatrelvir/ritonavir because it is a drug that can be used on an outpatient basis or in the case of a diagnosis of COVID-19 in a patient who was initially hospitalized for reasons other than infection (for example in the case of a nosocomial infection).
Comments 6: The safety assessment of vaccines includes a detailed enumeration of adverse reactions especially the thromboembolic and myocarditis adverse events. The risk–benefit evaluation for particular subgroups including young males and elderly patients with multimorbid conditions should include exact absolute risk numbers (if available).
Response 6: Data on the occurrence of the listed complications are described in paragraph 6 of the text (Safety and effectiveness of vaccinations against COVID-19). Additionally, we repeat this information in the added fragment of the summary.
Comments 7: A discussion of public health communication strategies to address vaccine hesitancy among patients with cardiovascular comorbidities would enhance the review.
Response 7: This issue is of course very important, but in our opinion it deserves a separate article. The primary goal of this narrative review is to increase awareness among physicians treating patients with cardiovascular diseases.
Comments 8: The manuscript emphasizes its focus on Polish data yet needs to enhance its discussion by clearly addressing differences in healthcare access and SARS-CoV-2 variant distribution between regions.
Response 8: As we explain in point 7 - awareness of the problem is key here. A detailed analysis of epidemiological trends with dynamically changing access to new versions of vaccines and updated recommendations, in our opinion, may contribute to blurring the main message of the text. We want to emphasize the validity of active prevention of COVID-19 in cardiology patients
Comments 9: A section should explicitly detail the review's limitations by addressing data gaps together with publication bias and difficulties in establishing cause-effect relationships between post-infection and post-vaccination events.
Response 9: This note is also included in the additional "Limitations" paragraph.
Minor Observations
Comments 10: The document contains minor typographical errors and inconsistent punctuation usage mainly affecting tabular data by using commas instead of periods for numeric values.
Comments 11: Long sentences appear in the document which could become easier to read by making them shorter through syntactic simplification.
Response 10 and 11: We have once again gone through the entire text for inaccuracies and linguistic awkwardness. We hope that you will find the corrections made sufficient.
Comments 12: The paper references Figures 1 and 2 correctly yet their legends require clearer explanations which should define abbreviations and clinical terms for better understanding.
Response 12: We are not sure about the intentions of this comment. Figures 1 and 2 were intended by the author to be as intuitive and "iconic" as possible. The limitation of the written text is intentional. We hope you agree with our version of visualization.
Comments 13: The meta-analytic results and vaccine safety data tables provide essential value to the study yet their interpretation could become more straightforward with additional explanatory footnotes.
Response 13: With all due respect, I do not understand this comment. I would appreciate additional explanation of this comment - what specific additions are missing? Perhaps the changes introduced in response to the comments of 6 reviewers have already met the requirements of this comment.
Comments 14: The reference list covers a wide range of sources and maintains currency. The accessibility of URLs is limited because some links do not provide complete access which may prevent peer verification at this time. Some references would benefit from DOIs instead of URLs when available.
Response 14: The references have been corrected.
Comments 15: More specific clinical recommendations especially for booster vaccinations and antiviral therapy timing would benefit from alignment with the latest international consensus statements as well as a brief discussion of differences among major societies (ESC, AHA, WHO, etc.).
Response 15: We have supplemented the text with as specific details as possible in the "Conclusion" section. “The situation is dynamic, as recommendations regarding the type of vaccines used may change each season, therefore it is probably best to respond in general terms and refer to current national and international recommendations”. This provision makes the conclusion universal and "resistant" to the dynamically changing detailed recommendations of various societies.
Dear Reviewer
Thank you once again for all your comments. We are convinced that their analysis and the corrections introduced have improved the quality of our work.
We also hope that you will accept our explanations and additions and that you consider our material worthy of publication.
Best regards
Authors
Reviewer 5 Report
Comments and Suggestions for Authors
The review written by Marcin Wełnicki et al., entitled "The Impact of COVID-19 and the Practical Importance of Vaccinations and Nirmatrelvir/Ritonavir for Patients with Cardiovascular Disease" is well presented and organized. It resume all information and results presented by previous articles. 80 recent articles were used to prepare the review. Sections are well presented and detailed. They englobe all topics regarding the impact of the COVID-19 antiviral treatments and vaccinations for patients with cardiovascular diseases. The main information highlighted in this review is that cardiologists should recommend booster doses of COVID-19 vaccines for their patients. In addition, the use of nirmatrelvir/ritonavir as antiviral therapy should be considered for cardiovascular patients because this therapy has the potential to reduce the risk of hospitalization and, in the case of high risk patients, to reduce mortality. This conclusion and information is with a high scientific sound especially for clinicians...
I recommend the publication of this review in its actual form.
Author Response
Dear Reviewer Thank you very much for such a positive opinion about our work.We are glad that our belief in the need to promote knowledge about vaccination and COVID-19
therapy among cardiologists has found your appreciation. With respect On behalf of the authors Marcin Wełnicki
Reviewer 6 Report
Comments and Suggestions for Authors
The authors provided a review article on the impact of COVID-19 on patients with cardiovascular diseases and emphasized the importance of vaccinations and antiviral treatments like Nirmatrelvir/Ritonavir for managing risks in this vulnerable group. The authors pointed out that vaccinations are highlighted as a safe and effective way to prevent serious infections among these patients. For Nirmatrelvir/ritonavir therapy, careful review of cardiac pharmacotherapy due to potential drug interactions is needed. This review can help readers better understand the treatment and preventive strategies management for cardiovascular patients with COVID-19.
Here are some questions and suggestions:
- The issue of mRNA vaccine side effects and their cardiotoxicity in cardiovascular patients with COVID-19 needs to be discussed.
- It would be best to discuss the vaccine's safety profiles across different subgroups of cardiovascular patients.
Author Response
Dear Reviewer
Thank you very much for your review of our article. First of all, thank you for your generally positive assessment. Below we address your comments and describe the modifications we have made.
Comments 1: The issue of mRNA vaccine side effects and their cardiotoxicity in cardiovascular patients with COVID-19 needs to be discussed.
Response 1: Thank you for highlighting this issue. We have added additional comments to the last paragraph regarding the safety of different types of vaccines, also addressing again the issue of incidents of myocarditis and pericarditis after subsequent doses of the mRNA vaccine. We have also emphasized the need for further observations and research. We hope that you will find this sufficient to supplement this topic
Comments 2: It would be best to discuss the vaccine's safety profiles across different subgroups of cardiovascular patients.
Response 2: We understand the need for a deeper analysis of this problem, but we believe that the current key problem is the low awareness of cardiologists about vaccinations in general. Vaccination against many diseases, COVID-19, but also influenza or penumococcus, can indirectly prevent exacerbations of cardiovascular diseases and cardiovascular incidents. The primary goal of our narrative review is to provide data on the advisability of vaccination against COVID-19 in the population of patients with cardiovascular diseases. We honestly discuss the issue of adverse effects, emphasizing the dominant benefits associated with vaccination. We also emphasize the need for further research and observation. Perhaps in the future, analysis of individual subgroups of cardiology patients will allow for a precise selection of the treatment strategy. At the moment, however, we believe that a holistic approach is more practical.
At the same time, however, we have introduced many point changes in our document. They are in the nature of clarifications of several aspects concerning qualification for treatment or the safety of vaccinations. They are in accordance with the suggestions you have indicated.
In the end, once again thank you for your in-depth analysis of our paper. We appreciate your constructive criticism and hope that the above clarifications and corrections will be considered sufficient.
Best regards,
Authors
Round 2
Reviewer 2 Report
Comments and Suggestions for Authors
The authors have addressed all my suggestions, and the manuscript is now in good shape for acceptance in its current form.